# Development of phage-containing hydrogel for treating *Enterococcus faecalis*-infected wounds

**Sahar Abed[1,2], Masoumeh Beig[2], Seyed Mahmoud Barzi[2], Morvarid Shafiei**[2]*****,
**Abdolrazagh Hashemi Shahraki[3], Sara Sadeghi[4], Aria Sohrabi[5]**

**1** Department of Microbial Biotechnology, Faculty of Basic Sciences and Advanced Technologies in Biology, University of Science and Culture, Tehran, Iran, **2** Department of Bacteriology, Pasteur Institute of Iran, Tehran, Iran, **3** Division of Pulmonary, Critical Care and Sleep, College of Medicine-Jacksonville, University of Florida, Gainesville, Florida, United States of America, **4** Department of Biological Sciences, Idaho State University, Pocatello, Idaho, United States of America, **5** Department of Epidemiology and Biostatics, Research Centre for Emerging and Reemerging Infectious Diseases, Pasteur Institute of Iran, Tehran, Iran

* dr.m.shafiei@pasteur.ac.ir

**Data Availability Statement:** All relevant data are within the manuscript itself. No additional supporting information files are provided.

## Abstract

### Background

Chronic wound infections caused by *Enterococcus faecalis* pose formidable challenges in clinical management, exacerbated by the emergence of vancomycin-resistant strains. Phage therapy offers a targeted approach but encounters delivery hurdles. Due to their biocompatibility and controlled release properties, hydrogels hold promise as carriers.

### Objective

This study aimed to fabricate phage-containing hydrogels using sodium alginate (SA), carboxymethyl cellulose (CMC), and hyaluronic acid (HA) to treat *E. faecalis*-infected wounds. We assessed the efficacy of these hydrogels both *in vitro* and *in vivo*.

### Methods

The hydrogel was prepared using SA-CMC-HA polymers. Phage SAM-E.f 12 was incorporated into the SA-CMC-HA hydrogel. The hydrogel's swelling index was measured after 24 h, and degradation was assessed over seven days. Surface morphology and composition were analyzed using Scanning Electron Microscopy (SEM) and Fourier-transform infrared spectroscopy (FTIR). Antibacterial activity was tested via optical density (OD) and disk diffusion assays. Phage release and stability were evaluated over a month. *In vivo* efficacy was tested in mice through wound healing and bacterial count assays, with histopathological analysis.

### Results

Hydrogels exhibited a swelling index of 0.43, a water absorption rate of %30, and 23% degradation over seven days. FTIR confirmed successful polymer incorporation. *In vitro* studies

**Funding:** The author(s) received no specific funding for this work.

**Competing interests:** The authors have declared that no competing interests exist.

demonstrated that phage-containing hydrogels significantly inhibited bacterial growth, with an OD of 0.3 compared to 1.1 for the controls. Hydrogels remained stable for four weeks. *In vivo*, phage-containing hydrogels reduced bacterial load and enhanced wound healing, as shown by improved epithelialization and tissue restoration.

## Conclusion

Phage-containing hydrogels effectively treat wounds infected with *E. faecalis*-infected wounds, promoting wound healing through controlled phage release. These hydrogels can improve clinical outcomes in the treatment of infected wounds.

## 1 Introduction

Chronic wound infections caused by *Enterococcus faecalis* pose a formidable challenge for clinical treatment. The emergence of vancomycin-resistant enterococci further complicates this challenge, severely limiting therapeutic options and increasing the risk of severe infections [1, 2]. Nowadays, phages are a promising treatment against antibiotic-resistant bacteria that can cause significant diseases in the environment, industry, and hospitals [3, 4]. In response to this pressing clinical need, alternative therapeutic approaches, such as phage therapy, have gained attention for their ability to offer targeted infection control without disrupting the beneficial microbiota [3, 4].

Delayed wound healing is a multifaceted process influenced significantly by the proliferation of pathogens, which can lead to chronic infections and hinder the body's natural healing mechanisms.

The proliferation of pathogens in wound environments leads to delayed healing through several mechanisms, including forming biofilms, excessive inflammatory responses, impaired cellular migration, and proliferation. The pH of the wound environment also plays a significant role in healing dynamics. An acidic environment benefits wound healing, while alkaline conditions can be associated with chronic wounds and delayed healing [5, 6]. This is particularly relevant in infection, as many pathogens thrive in alkaline conditions, further complicating the healing process. The balance between maintaining a conducive pH and controlling pathogen proliferation is thus crucial for effective wound management [6]. Addressing these factors is essential for promoting effective wound healing and preventing chronic infections [7–9].

Phage therapy offers distinct advantages over traditional antibacterial agents, particularly in treating infections resistant to conventional antibiotics. Due to rising antibiotic resistance, traditional antibiotics are becoming less effective, making phage therapy a viable alternative.

On the other hand, the lack of horizontal gene transfer of phages significantly reduces the potential for disseminating antimicrobial resistance genes among bacterial populations. Phages offer high specificity, targeting only specific bacterial strains without harming the surrounding healthy tissues or the standard microbiota [10]. This specificity helps prevent secondary infections and reduces collateral damage [10]. Additionally, phages can disrupt biofilms, are often antibiotic-resistant, and contribute to chronic wounds [11]. Phage therapy has also been shown to promote wound healing more effectively than traditional antibiotics by enhancing the expression of healing proteins and facilitating tissue repair [12]. Another advantage of phages is their adaptive capability, making it challenging for bacteria to develop resistance against them [13]. Moreover, the topical administration of phage therapy allows for localized treatment, minimizing systemic side effects and maximizing therapeutic efficacy at

the infection site [12]. These attributes underline the advantages of phage therapy as a potent alternative to conventional antibacterial agents in wound care.

The mechanism of action of phages against bacterial infections encompasses specific binding to bacterial receptors, injection of genetic material, replication within the host, production of lytic enzymes, and subsequent bacterial lysis [14, 15]. This multifaceted approach, combined with the ability to enhance immune responses and target biofilms, positions phage therapy as a potent alternative to conventional antibiotics in the fight against multidrug-resistant bacterial infections.

Phages, known for their specificity and potential efficacy against *E. faecalis*, have emerged as promising agents [16]. However, challenges persist, including the need for stability and sustained release at infection sites, which currently hinder the clinical application of phages [6]. Currently, phage therapy for wound care is typically administered by directly applying phages to the wound site. Although traditional methods like bandages, gauzes, and ointments have been considered to enhance this treatment, they need to act as effective carriers for phages due to their inability to sustain phage release and maintain viability. Integrating specialized carriers, such as hydrogels, significantly advances this field.

Hydrogels have emerged as preferred delivery systems due to their exceptional biocompatibility, moisture-retention capabilities, and ability to provide controlled and sustained release of therapeutic agents [17–19]. Incorporating phages into a hydrogel matrix ensures prolonged contact with infected tissues, effectively enhancing the antibacterial efficacy and improving the overall effectiveness of phage therapy in wound care management [20]. These natural polymers are preferred over synthetic polymers because of their abundance, non-toxicity, and biodegradability [21]. Numerous recent studies have explored using hydrogels as carriers to deliver antibacterial compounds. Despite recent studies, substantial gaps still need to be in understanding the full potential of phage-containing hydrogels for treating *E. faecalis*-infected wounds. This study aimed to bridge these gaps by comprehensively reviewing the progress in phage therapy and hydrogel delivery systems.

Recently, we characterized SAM-E.f 12, a novel lytic phage that has potent activity against *E. faecalis* [22]. Building on this foundation, the present study aimed to innovate in the field of wound management by formulating a new hydrogel that not only incorporates this novel phage and develops phage-containing hydrogel but also leverages the unique properties of hyaluronic acid (HA). Unlike traditional hydrogel formulations, our hydrogel utilizes HA, a natural polymer known for its exceptional skin repair capabilities and biocompatibility. It has not been previously applied in this context for wound healing. By integrating HA with sodium alginate (SA) and carboxymethyl cellulose (CMC), we developed a hydrogel capable of delivering SAM-E.f 12 directly to infected wound sites, which can combine antimicrobial action with enhanced tissue repair. Finally, we aimed to rigorously evaluate our phage-containing hydrogel antimicrobial properties and wound healing capabilities *in vitro* and *in vivo* using a mouse model infected with *E. faecalis*.

## 2 Methods and materials

### 2.1 Preparation of hydrogel and load with phage

To prepare the hydrogel, 4 g of SA (Merck, Germany) and 0.05 g of HA (Merck, Germany) were gradually dissolved in 100 ml of distilled water at 40 degrees with continuous stirring to obtain a homogeneous solution. Concurrently, at room temperature, 0.2 g of CMC (ACROS, Dessel, Belgium) was slowly dissolved in 10 mL distilled water. Subsequently, two solutions were combined, autoclaved, and cooled to create a hydrogel solution ready for further application. After sterilization and cooling, the solution was used to form a hydrogel. Subsequently,

200 µL of the phage solution (SAM-E.f 12) was meticulously mixed with the hydrogel solution for 10 min using a magnetic stirrer to ensure a uniform distribution of the phage. One ml sterile 1% Calcium chloride ($CaCl_2$) Merck (Darmstadt, Germany) solution was added using a syringe to induce gel formation. Following the cross-linking process, 200 µL of the SAM-E.f 12 phage solution, previously isolated and characterized in our study [22], was incorporated into the hydrogel. The resulting solution contained the SAM-E. f 12 phage and $CaCl_2$ were poured onto a plate to form a film, which was left to solidify at—20˚C for 18–24 h. The solidified hydrogel film was cross-linked by adding 8 mL of 10% $CaCl_2$ solution onto its surface. After one hour, the excess liquid was removed.

## 2.2 Characterization of the SA-CMC-HA hydrogels

**2.2.1 Swelling index evaluation.**   We initiated our investigation by cutting the hydrogel film into pieces measuring $2 \times 2$ cm [23]. These pieces were then allowed to air dry at ambient room temperature until scorched. After drying, the initial weight of each piece ($W_0$) was recorded. Subsequently, they were submerged in distilled water and incubated at 37˚C for seven days.

The weight of the hydrogel was measured (Ws) after 24 h and subsequently daily for one week. This equation was used to calculate the water absorption rate and the swelling ratio [24].

$$\text{Water absorption rate } (\%) = [(W_s - W_0)/W_s] \times 100$$

$$\text{Swelling ratio } = (W_s - W_0)/W_0$$

**2.2.2 Hydrogel degradation.**   To assess the degradation of the hydrogel, the following procedure was employed: A dry hydrogel sample of $2 \times 2$ cm was initially weighed ($W_0$). This sample was then submerged in 5 mL of Phosphate-buffered saline (Sigma, Aldrich) and maintained at 37˚C. At predetermined intervals, the hydrogel was retrieved, rinsed with distilled water, and reweighed on days 1, 3, 5, and 7 (Wt). The formula provided was utilized to compute the degradation of hydrogel over time [25]: Remaining weight (%) = $(Wt/W_0) \times 100$

**2.2.3 Scanning Electron Microscopy (SEM) of hydrogels.**   The surface morphology and overall structure of the control and phage-containing hydrogels were examined using SEM. Before analysis, the hydrogels were thoroughly dried and coated with multiple thin metal layers, such as gold. The SEM analysis used a Tescan-10 KV instrument [26].

**2.2.4 Fourier-Transform Infrared Spectroscopy (FTIR).**   The SA-CMC-HA hydrogel and its constituent parts, including SA, CMC, and HA powder, underwent FTIR spectrum analysis using a Perkin-Elmer FTIR model 2000 spectrometer, covering the range of 400–4000 $cm^{-1}$ to ascertain their chemical composition and bond structure. The resulting spectra were examined to identify the characteristic absorption bands corresponding to the vibrational states of the functional groups in the hydrogel [27].

## 2.3 *In vitro* antibacterial activity assay

**2.3.1 Phage release assay.**   Three tubes containing 5 mL of Brain Heart Infusion (BHI) broth (Merck, Germany) were inoculated with a single colony of *E. faecalis*, and the bacterial suspensions were prepared using a sterile loop. Control Tube: The first tube was designated as the control and contained only the bacterial suspension without any added hydrogel or phage treatment. Phage-containing hydrogel Tube: The second tube added a phage-containing hydrogel to the second tube to evaluate its effectiveness in delivering and releasing phages. Direct Phage Tube: The third tube received a direct addition of 20 µL of phage stock solution to assess the efficacy of the phage application in the absence of a hydrogel carrier.

Optical density (OD) measurements assessed the phage release [28]. This procedure was conducted to evaluate the antimicrobial effects of the hydrogels and observe the phage release dynamics. The release of SAM-E.f 12 phage from the hydrogel was investigated by placing a phage-containing hydrogel sample in a sterile culture tube containing BHI broth and *E. faecalis*. The tubes were incubated at 37˚C, and the OD was subsequently measured using a spectrophotometer.

**2.3.2 Disk diffusion assay.** The antimicrobial efficacy of the SA-CMC-HA hydrogels was examined using the disc diffusion method [29]. Bacterial cultures prepared overnight and standardized to a 0.5 McFarland density were inoculated onto BHI agar plates with a sterile swab.

Disks with a diameter of 3 mm were cut from phage-containing hydrogel and hydrogel without phages (negative controls). These discs were assigned to a bacterial lawn, and the plates were incubated at 37˚C for 24 hours. Growth inhibition around the discs was observed.

## 2.4 Stability studies

In this study, the primary focus was on assessing the stability of the SAM-E.f 12 phage at 25˚C within the hydrogel system. Weekly evaluations were conducted over a month-long storage period to monitor phage viability and count. For biological stability assessments, a 2 x 2 cm hydrogel sample embedded with the phage was stored in SM buffer (150 mM NaCl (Merck, KGaA, Darmstadt, Germany), (40 mM Tris-HCl (pH 7.5) (Fisher Scientific F, Loughborough, UK), and ten mM MgSO (Merck, KGaA, Darmstadt, Germany) at 25˚C.

While this study focused on *in vitro* conditions, it is essential to note that the stability of the SAM-E.f 12 phage under various physiological conditions, including pH levels (ranging from 2 to 14), temperature extremes (-20, 4, 37, 50, 60, and 70˚C), and varying saline concentrations (5%, 10%, and 15% NaCl at 37˚C), has been previously investigated [22].

**2.4.1 Plaque assay.** To accurately assess the phage count at 0, 1, 2, 3, and 4 weeks, we employed double-layer agar for conducting plaque assay, a variant of the traditional plaque assay method that allows for rapid and efficient determination of phage concentration in a sample [30, 31].

## 2.5 *In vivo* studies

Animal experiments were performed under code IR according to the instructions of the Pasteur Institute of Iran Ethics Committee, Department of Bacteriology. IR.PII.AEC.1403.008.

Male BALB/c mice, aged five weeks, weighing 25 ± 2 g, were obtained from the Pasteur Institute. The mice were acclimated to their new environment for one week. Mice were divided into six groups, each containing five mice.

I. Healthy Control: Healthy, untreated mice.

II. Wounded Negative Control: Wounded mice were treated with sterile distilled water.

III. Positive Control: Wounded, infected, and untreated mice

IV. Phage Hydrogel Group: Wounded infected mice treated with phage-containing hydrogels.

V. Hydrogel Without Phage: Wounded, infected mice treated with plain hydrogel.

VI. Direct Phage Group: Wounded, infected mice treated directly with phages.

**2.5.1 Model for wound formation and infection.** The mice were administered an intraperitoneal injection of ketamine/xylazine (60 mg/kg) to induce anesthesia [32]. We shaved the

mice's back hair (between vertebrae L2-L6) with shaving cream to create wounds. The wound sites were sterilized before the procedure. To minimize bleeding, a five mm incision was made with a biopsy punch, entering the epidermis and superficial dermis without touching the muscles. The wound only affected the muscular tissue, reaching the maximum skin depth. Wounds were purposefully infected with 50 µl of a 1.5 x 10$^8$ CFU/ml *E. faecalis* solution without the negative control group.

**2.5.2 *In vivo* assessment of hydrogels' antibacterial efficacy.** After 24 h of wound formation, all wounds, except those in the positive control group, were treated with hydrogel without phage, phage-containing hydrogel, and direct phage.

After 24 hours, we treated all wounds, except the positive control group, with either hydrogel without phages, phage-containing hydrogel, or direct phage application. On days 1, 3, 7, 10, and 14 of the post-treatment, wound samples were collected to evaluate the effectiveness of the hydrogels in fighting bacteria. Each sample was obtained by gently swabbing the wound site with a sterile swab moistened with 2 mL of sterile distilled water prepared explicitly for the lesion. The swab was then placed in a tube containing 3 mL of sterile physiological saline. After thorough mixing, 100 µL of the solution was cultured on blood agar (Merck, Germany) plates. The plates were incubated at 37˚C for 24 hours, after which the number of colonies was enumerated.

**2.5.3 Wound healing evaluation.** To assess the wound healing process, the wounds on each mouse were photographed, and their diameters were measured on days 1, 3, 7, 10, and 14 post-wounding. Digital calipers were used to accurately measure the wound diameters, ensuring consistent measurements across all observations. Wound contraction, indicative of healing progress, was quantified using the following equation to calculate the percentage of wound size reduction:

$$\text{Wound contraction } (\%) = (A0 \; - \; AT) \times 100$$

A0 represents the initial wound area immediately after creation, and AT represents the wound area on different days of treatment.

**2.5.4 Histopathological evaluation.** Histopathological analysis was performed as previously described in [22]. We used a comprehensive scoring system [33] to compare histopathological changes among several experimental groups. This systematic approach enhanced the accuracy of the analysis and provided detailed insights into the alterations observed under different testing conditions.

# 3 Results

## 3.1 Preparation of hydrogels

In this study, we designed a hydrogel using natural polymers SA, CMC, and HA cross-linked with 10% CaCl$_2$. To prepare the hydrogel, SAM-E. f 12 phages were introduced to infuse phages. Control samples were used without phages (Fig 1).

**3.1.1 Confirmation of CaCl2 crosslinking with SA-CMC-HA polymers by visual observation.** Visual observation, which involves examining changes in the physical appearance of the hydrogel before and after the addition of CaCl2, was one of the initial and simplest methods for confirming hydrogel crosslinking. Crosslinked hydrogels typically transition from a liquid or semi-liquid state to a more solid gel form. Before adding the cross-linked, the hydrogel is, but after adding the cross-linked, have the solid.

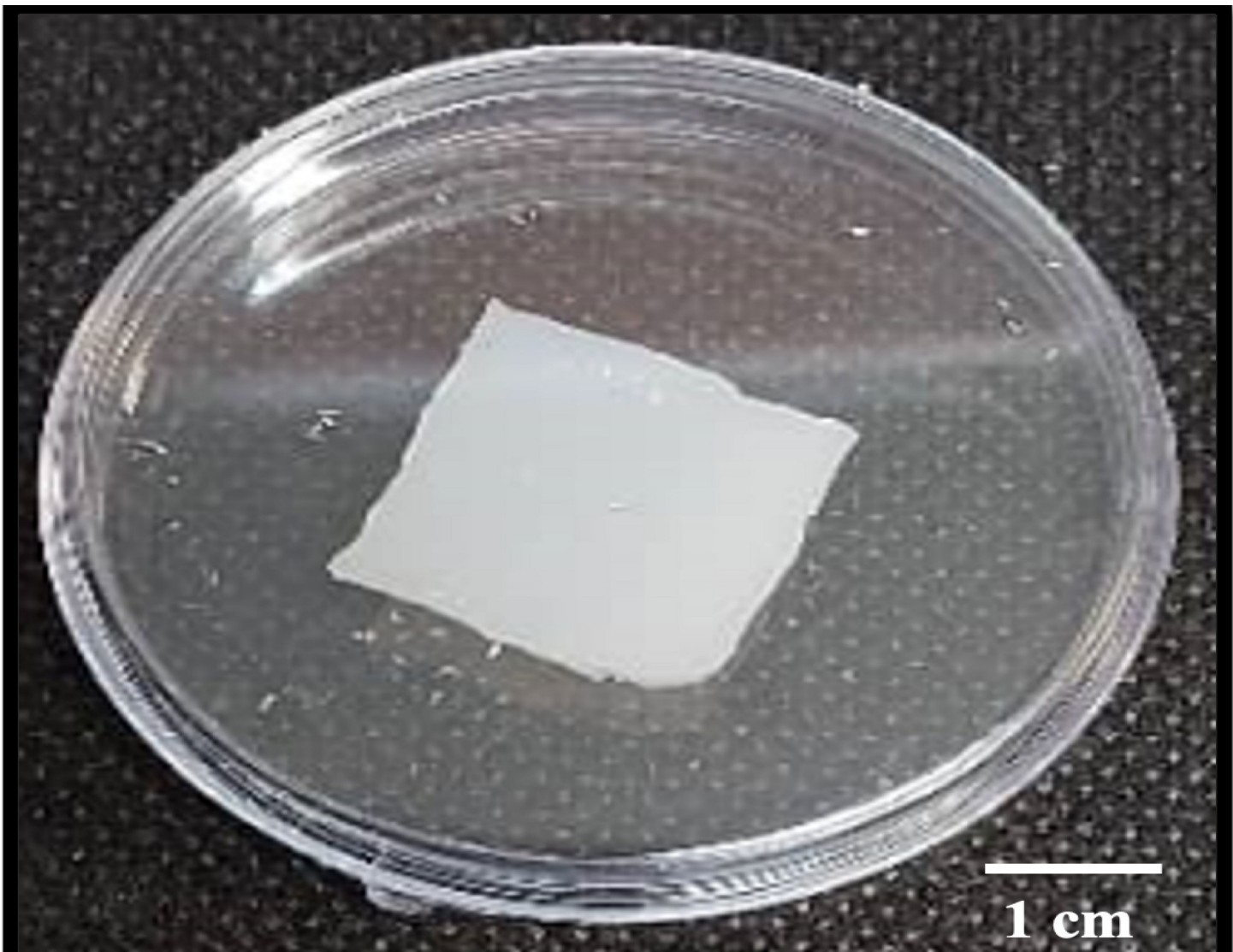

**Fig 1. Formation of hydrogel film.** This figure shows the hydrogel film placed within a petri dish, demonstrating its uniform structure and translucent appearance post-synthesis.

## 3.2 Characterization of the SA-CMC-HA hydrogels

**3.2.1 Swelling index evaluation.** After immersion for seven days, the SA-CMC-HA hydrogel exhibited a swelling index of 0.43 and a water absorption rate of %30 (Fig 2A and 2B).

**3.2.2 Degradation behavior of hydrogel.** After seven days, the SA-CMC-HA hydrogel degraded, losing approximately 23% of its weight (Fig 3).

**3.2.3 Surface morphology analysis of hydrogel using SEM.** A voltage of 10 kV was used to investigate the surface morphologies of the pure hydrogel and the SAM-E.f 12 phage-injected hydrogel [23]. The pure hydrogel has a smooth surface (Fig 4A), whereas the phage-injected hydrogel exhibited minor irregularities (Fig 4B) caused by phages. On average, the bump diameter is approximately 600 nm. This suggests that hydrogels may be suitable for phage encapsulation (Fig 4A and 4B).

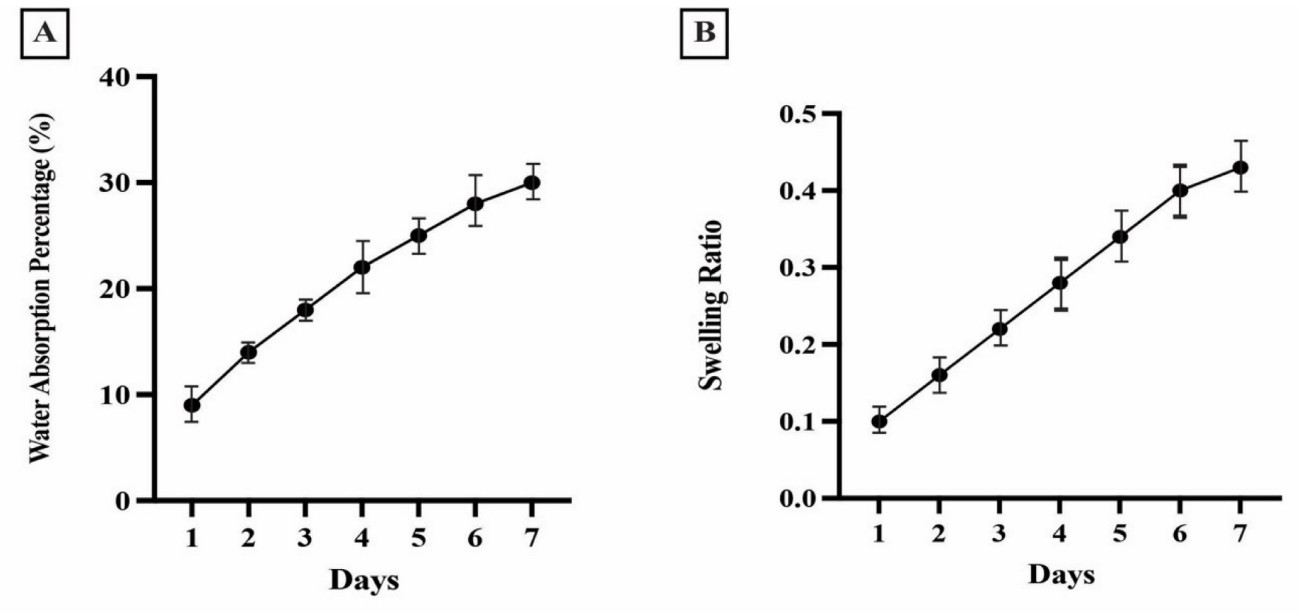

**Fig 2. Characterization of SA-CMC-HA hydrogels. (A)** Water Absorption Rate in SA-CMC-HA Hydrogel In 7 days. **(B)** Swelling Ratio in SA-CMC-HA Hydrogel In 7 days.

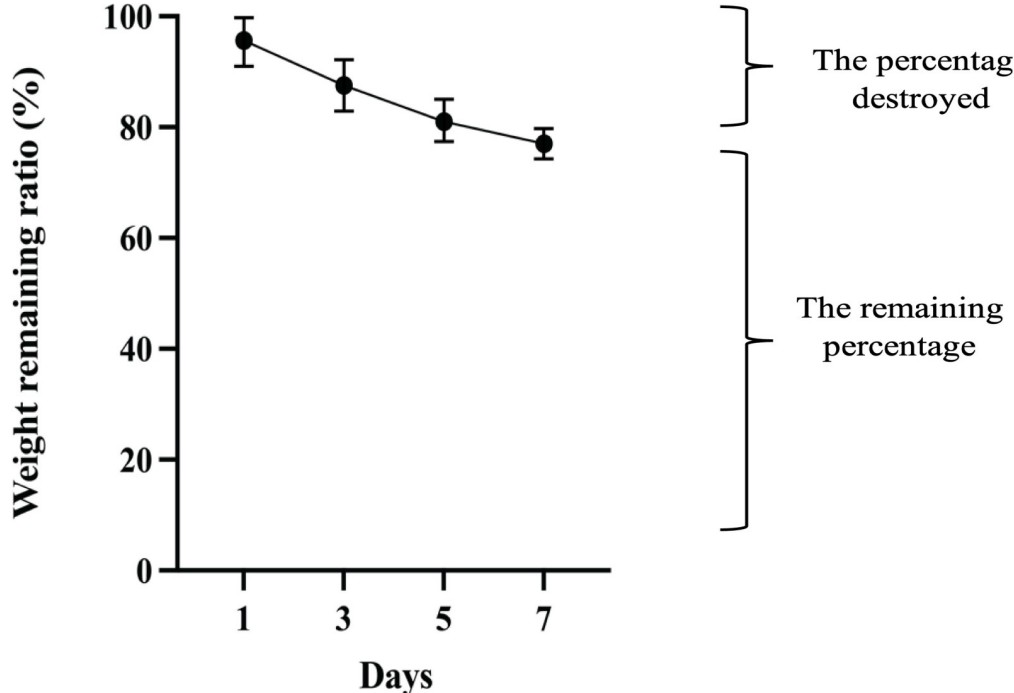

**Fig 3. Degradation profile of SA-CMC-HA hydrogel over seven days.** The figure presents the degradation behavior of the SA-CMC-HA hydrogel over a period of seven days, measured as the percentage of hydrogel mass remaining (%). The degradation shows a gradual decline in hydrogel mass over time, which decreases steadily until Day 7. This profile indicates the hydrogel's stability and controlled biodegradability, suitable for applications requiring a slow degradation rate.

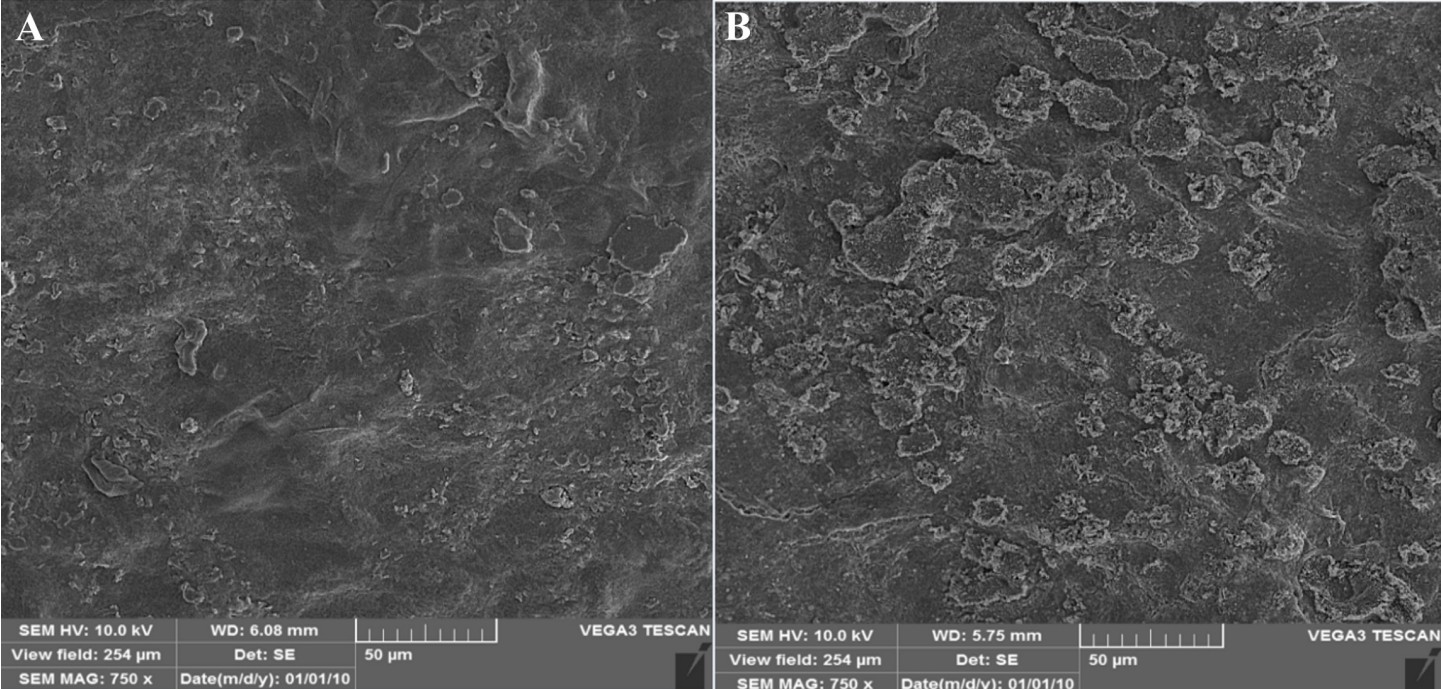

**Fig 4. Surface morphology at a scale of 50 μm.** (A): Control hydrogel with an almost uniform and smooth surface. (B): Hydrogel infused with bacteriophage SAM-E.f 12 exhibiting a protruding surface texture.

### 3.2.4 Analysis of hydrogel components and SA-CMC-HA hydrogel using FTIR spectroscopy

FTIR spectroscopy was employed to investigate the chemical composition and bond structure of the SA-CMC-HA hydrogel and its components (Fig 5). Analysis of the FTIR spectra of SA, CMC, HA, and the final SA-CMC-HA hydrogel provided valuable insights into the presence of functional groups in each sample.

The functional groups (C = O and potentially C-N) in the SA-CMC-HA hydrogel spectrum (Fig 5D) indicate the successful incorporation of SA, CMC, and HA into the final hydrogel structure. The broad peak around 3200–3500 cm$^{-1}$ confirms the presence of hydroxyl groups (OH) from all three components. In the 1620–1420 cm$^{-1}$ region, the hydrogel spectrum showed a combination of peaks potentially arising from C = O stretching in carboxylic acids (from SA and CMC) (Fig 5A and 5B) and amide groups (from HA) (Fig 5C). Although some overlap might be present, the distinct amide I and amide II peaks characteristic of HA are still somewhat observable, suggesting the presence of HA within the hydrogel (Fig 5A–5D).

## 3.3 *In vitro* antibacterial activity assay

**3.3.1 Phage release assay.** The OD measurements of the bacterial cultures after 24 h revealed that the direct application of SAM-E.f 12 phage and the phage-containing hydrogel significantly inhibited bacterial growth in the broth medium. Specifically, the OD was 0.38 for the phage-containing hydrogel and 0.32 for the direct phage application, demonstrating the effective release and antimicrobial activity of phages from the hydrogel. In contrast, the control group showed a significant increase in turbidity and OD (1.10) after 24 hours, indicating substantial bacterial growth without phage treatment (Fig 6A and 6B).

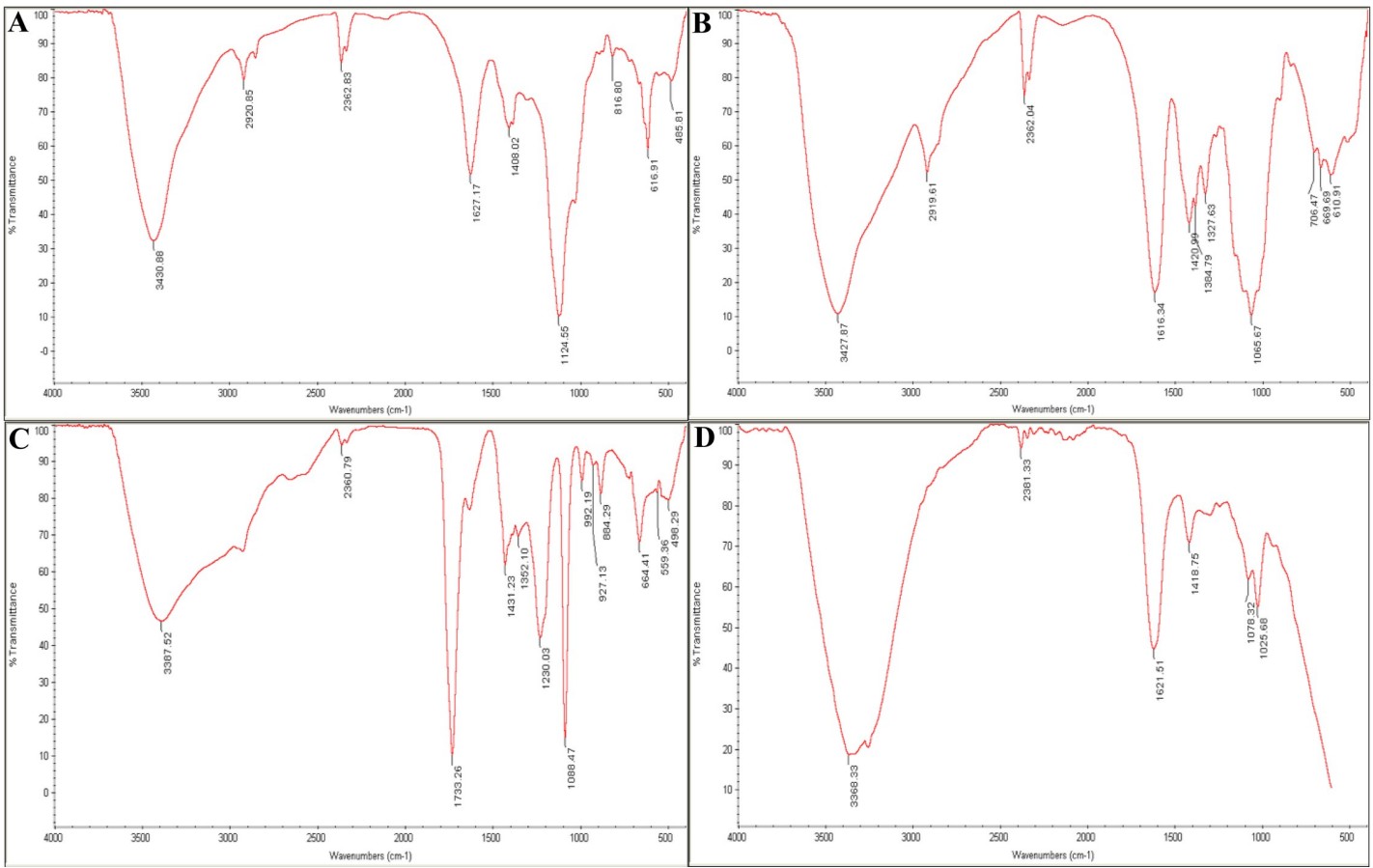

**Fig 5.** FTIR spectrum of (A). SA, (B). CMC, (C). HA, and (D). SA-CMC-HA Hydrogel. The peaks in the range of 3200–3500 cm-1 are related to (O-H), those in the 1620–1420 cm-1 range to (C = O and C-N) bonds, and those at 1020 cm-1 to (C-O) bonds. The presence of distinctive peaks at 2330 cm$^1$ in the spectra of the phage-loaded hydrogel indicates the possible retention of phages within the hydrogel.

**3.3.2 Disk diffusion assay.** The disk diffusion assay demonstrated the efficacy of the phage-containing hydrogel in inhibiting bacterial growth. A clear zone of inhibition measuring 32 mm was observed around the hydrogel-containing phage disc (Fig 7A), indicating its potent antibacterial activity. In contrast, the blank hydrogel disc used as a negative control did not produce any observable inhibition zone, confirming that the hydrogel alone did not affect bacterial growth (Fig 7B). Moreover, the direct application of phage resulted in a distinct inhibition zone measuring 29 mm around the bacterial colonies (Fig 7C), further validating the antibacterial potency of the phage in both direct and hydrogel-encapsulated forms.

## 3.4 Storage stability of optimized phage-containing hydrogel

Phages must remain stable and maintain their infectivity during storage to ensure commercial viability. In this study, we evaluated the stability of the SAM-E.f 12 phage within the SA-CMC-HA hydrogel over four weeks at 25˚C.

**3.4.1 Plaque assay.** The phage count was monitored at regular intervals (0, 1, 2, 3, and 4 weeks) using the double-layer agar plaque assay (Fig 8).

At week 0, the phage count was recorded at 6.78 log PFU/mL. After one week of storage, the count slightly decreased to 6.50 log PFU/mL, followed by a gradual decline to 6.25 log

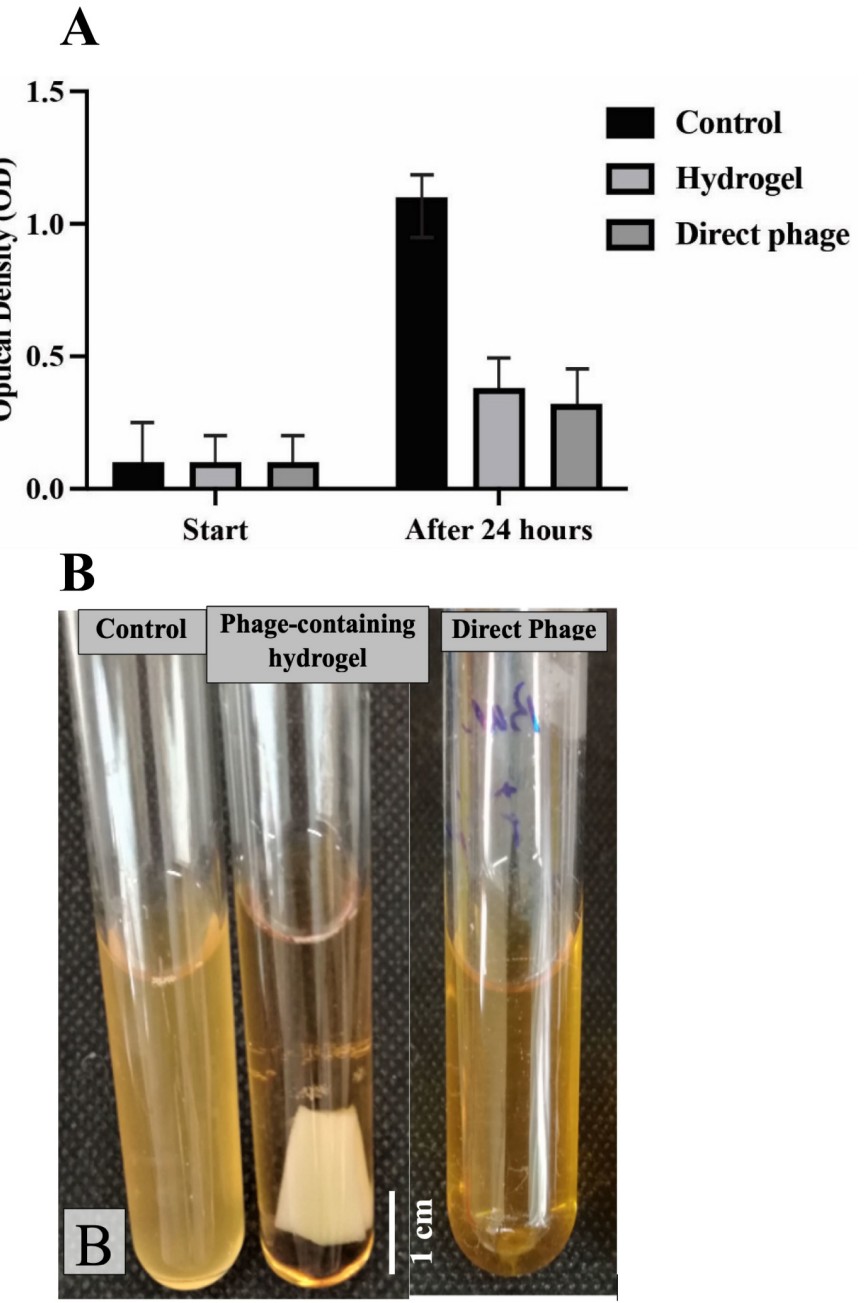

**Fig 6.** (A) This graph shows that the direct phage and phage-loaded hydrogel effectively suppressed bacterial growth, as evidenced by the steady decrease in optical density (OD) of the bacterial culture over time compared to the control. (B) BHI broth contains only Enterococcus faecalis (Control), phage-containing hydrogel, and direct phage. The effectiveness of the hydrogel-containing phage and direct phage in a liquid culture medium is illustrated, with the inhibitory effect contrasting sharply with the control group (no hydrogel), which showed a significant increase in OD.

PFU/mL at week 2. By week 3, the count was 6.16 log PFU/mL; by the end of the fourth week, the count reached 6.02 log PFU/mL.

Overall, the SAM-E.f 12 phage showed only a minor decrease in count over the four-week storage period, with a total reduction of 0.76 log PFU/mL. This suggests that the phage

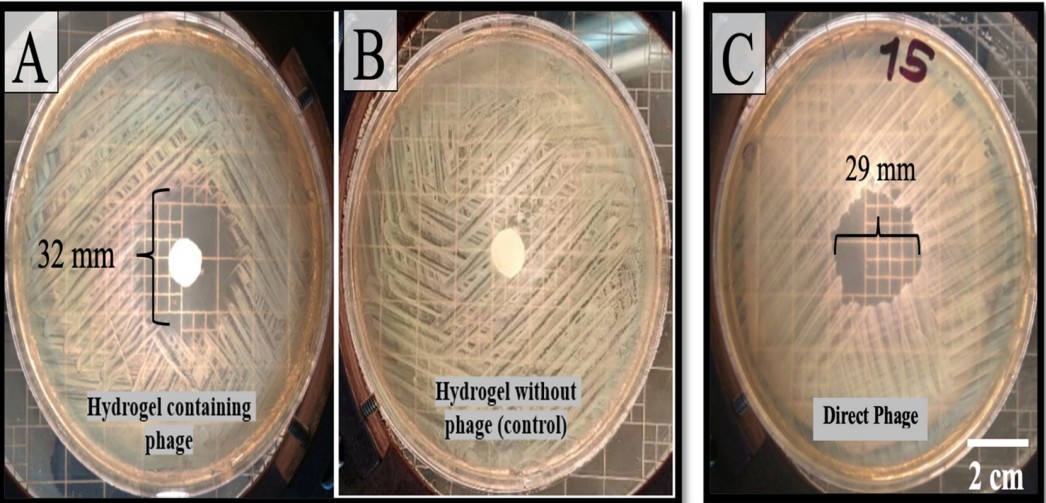

**Fig 7.** (A) A disk infused with hydrogel-containing phage produced a clear halo around the surrounding bacteria. (B) Blank hydrogel discs were used as negative controls, and no halo formation was observed. (C) Direct phage produced a clear halo around the surrounding bacteria.

remained relatively stable within the SA-CMC-HA hydrogel matrix, maintaining its infectivity under storage conditions for an extended period.

### 3.5 *In vivo* studies

**3.5.1 *In vivo* assessment of hydrogels' antibacterial efficacy.** The bacterial load in the wounds was assessed at various time intervals using swab cultures (Fig 9A). Scars were infected

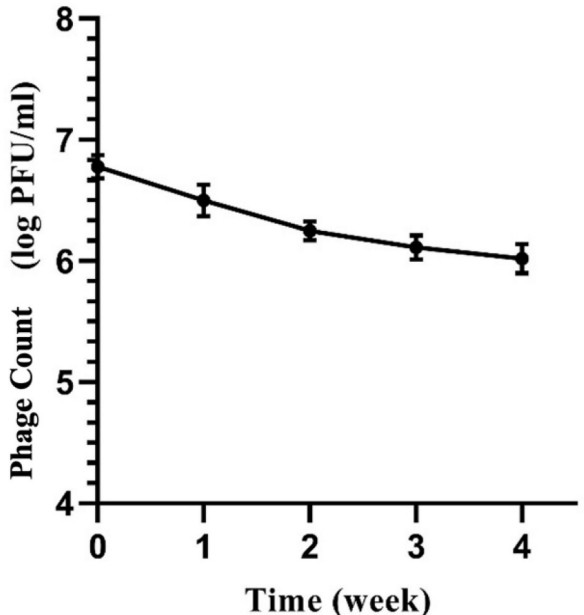

**Fig 8. The biostability of phage SAM-E.f 12 in SA-CMC-HA hydrogel was evaluated after storage for 0, 1, 2, 3, and 4 weeks at 25°C.** The phage count was determined using double-layer agar for conducting plaque assay (n = 3), and the error bars represent the standard deviations.

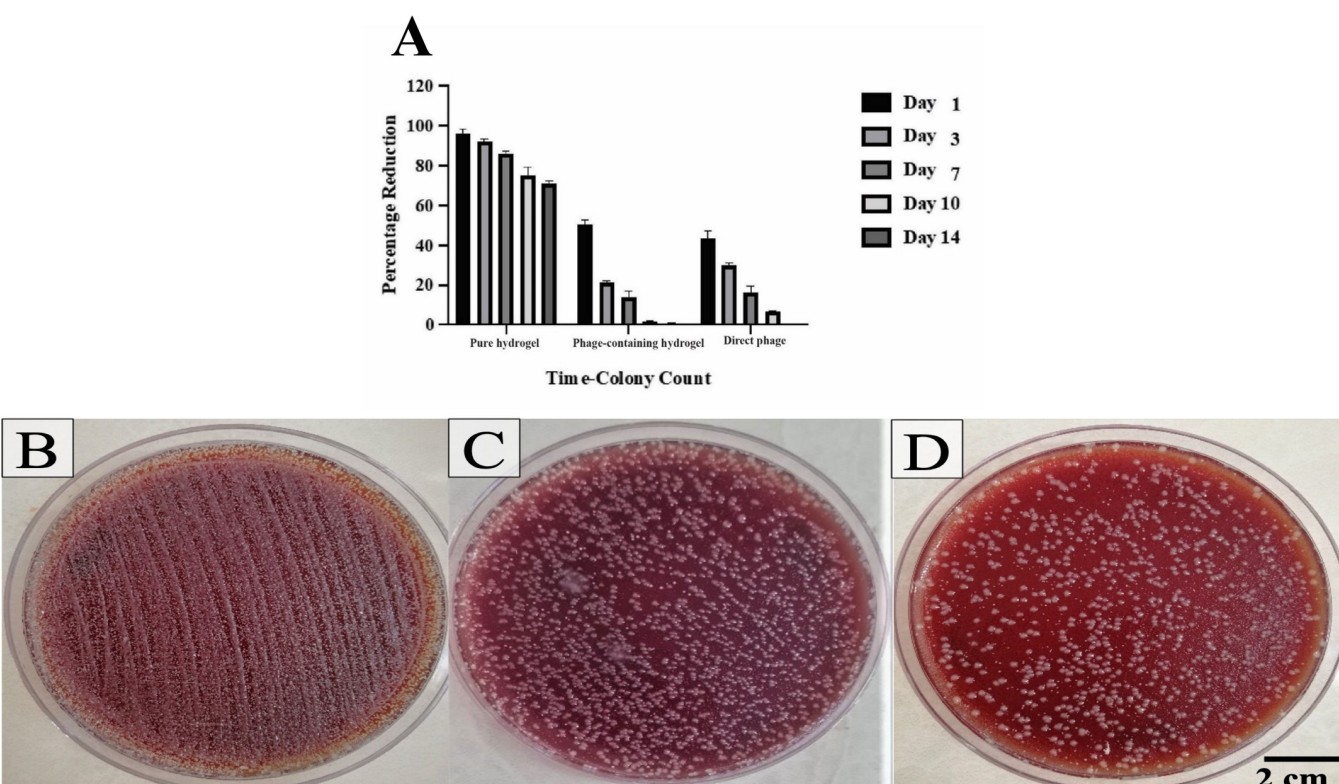

**Fig 9.** (A) Comparative Analysis of Treated Groups. The chart illustrates the bacterial load in different groups of mice on days 1, 3, 7, 10, and 14. Wound samples were collected at specific intervals. (B) The group treated with pure hydrogel (without phage), (C) The group treated with hydrogel containing phage, (D) The group treated with direct phage.

with *E. faecalis* on the first day after wound creation. Treatment was initiated 24 hours later, on the second day. The first wound sampling was conducted on the first day, followed by additional samplings on Days 3, 7, 10, and 14.

The group treated with pure hydrogel (without phages) showed the highest bacterial load, particularly on Day 3, where a significant bacterial presence was observed (Fig 9B). In contrast, groups treated with phage-containing hydrogel (Fig 9C) and those treated with direct phage (Fig 9D) demonstrated much lower bacterial counts at all time points.

Specifically, for the group treated with pure hydrogel, the percentage reduction in bacterial colony count was as follows: Day 1: 98%, Day 3: 93%, Day 7: 87%, Day 10: 77%, and Day 14: 71%. The phage-containing hydrogel group showed a more substantial reduction in bacterial load, with bacterial counts decreasing progressively from Day 1: 51%, Day 3: 22%, Day 7: 16%, Day 10: 2%, to a complete reduction on Day 14: 0%. Similarly, the group treated with direct phage exhibited reductions of Day 1: 40%, Day 3: 31%, Day 7: 19%, Day 10: 7%, and Day 14: 0%.

Additionally, all mice treated with the hydrogel-embedded phage survived and maintained excellent overall health. In contrast, 80% of the mice in the Positive Control group (III) and 60% in the group treated with pure hydrogel (V) did not survive the treatment. Blood cultures from the deceased mice confirmed the presence of *E. faecalis*, indicating that bacterial sepsis caused their death.

**3.5.2 Wound healing.** Phage-containing hydrogels were prepared for investigation in wound regeneration. Mice were divided into different groups, and their effectiveness in wound healing and epithelialization was evaluated. The group treated with the phage-

containing hydrogel showed the most effective healing, significantly improving on the third day and complete recovery on the 14th. In contrast, the group treated with hydrogel without phage showed delayed healing and poor recovery. The group treated with the direct phage showed wound healing, but compared to group (IV), there was less repair (Fig 10A).

Fig 10B shows the effect of the phage-containing hydrogel on wound healing in male BALB/c mice infected with *E. faecalis*. The wound healing rates were monitored over 14 days across different treatment groups, including the Wounded Negative Control (II), Positive Control (III), Phage-containing Hydrogel group (IV), Hydrogel without Phage (V), and Direct Phage (VI).

The Wounded Negative Control group (Group II) began with wound sizes of 5 mm on Day 1, which slightly reduced to 4.5 mm by Day 3, demonstrating a steady pace of natural healing. By Day 7, the size further decreased to 3 mm, and by Day 10, the wound had halved again to 1.5 mm. By the final observation on Day 14, the wound had nearly closed, measuring only 0.62 mm.

In contrast, the Positive Control group (Group III), which was infected but untreated, initially saw an increase in wound size to 6.3 mm by Day 3, indicating an aggravation of the infection. This size slightly decreased to 5.9 mm by Day 7 and 5.5 mm by Day 10, showing prolonged healing. By Day 14, only a modest reduction to 4.8 mm was noted, reflecting a struggle to overcome the infection naturally. The group treated with the Phage-containing Hydrogel (Group IV) saw an initial decrease in wound size to 4.2 mm by Day 3, suggesting an immediate antibacterial effect from the phage. The size rapidly declined to 2.7 mm by Day 7 and 1.1 mm by Day 10, nearing closure. By Day 14, the wound size had reduced dramatically to just 0.2 mm, indicating almost complete healing and successful management of the bacterial challenge.

The Hydrogel without Phage group (Group V) also demonstrated some wound closure, albeit slower, with a reduction to 5.5 mm by Day 3 and 4.9 mm by Day 7. By Day 10, the wound had reduced further to 4.4 mm. By Day 14, it measured 4 mm, showing that the hydrogel alone provided some protection and promoted healing but lacked the antibacterial properties necessary for more effective treatment. Finally, the group treated with Direct Phage (Group VI) displayed a reduction from 5 mm on Day 1 to 4.6 mm by Day 3, reflecting effective initial antibacterial action. The wound size decreased to 3.1 mm by Day 7 and 1.7 mm by Day 10. By Day 14, the wound had significantly healed, closing to 0.8 mm, demonstrating improved healing compared to the positive control, though slightly less effective than the phage-containing hydrogel. This comparative analysis highlights the potential of phage therapy in wound care, mainly when utilized within appropriate delivery matrices like hydrogels.

**3.5.3 Histopathological evaluation.** The histopathological analysis of tissue samples recovered from wounds on day 14 across different experimental groups revealed distinct tissue response patterns, highlighting the potential therapeutic effects of phage-containing hydrogel treatment (Table 1). A four-point grading system was utilized to assess various stages of wound healing. Tissue samples were evaluated based on the following criteria: cell density, oedema, inflammatory cell infiltration, fibrogenesis, angiogenesis, epithelialization, fibrosis, hair follicle presence, and epidermal thickness. Each parameter was scored as follows: (0) indicating normal histology with no alterations, (1) for mild alterations affecting less than 25%, (2) for moderate alterations affecting 25% to 50%, and (3) for severe alterations affecting more than 50% of the tissue. Compared to the positive control group, which exhibited intense fibroblast activity (score: 3), no epidermal layer formation (0), high infiltration of inflammatory cells (neutrophils) (3), no epithelization or hair follicle re-formation (0), significant fibrosis (3), and moderate angiogenesis (2) (indicative of severe inflammation and fibrosis), the phage-containing hydrogel-treated group showed more controlled tissue responses. This group demonstrated intense fibroblast activity (3), appropriate epidermal thickness (3), reduced

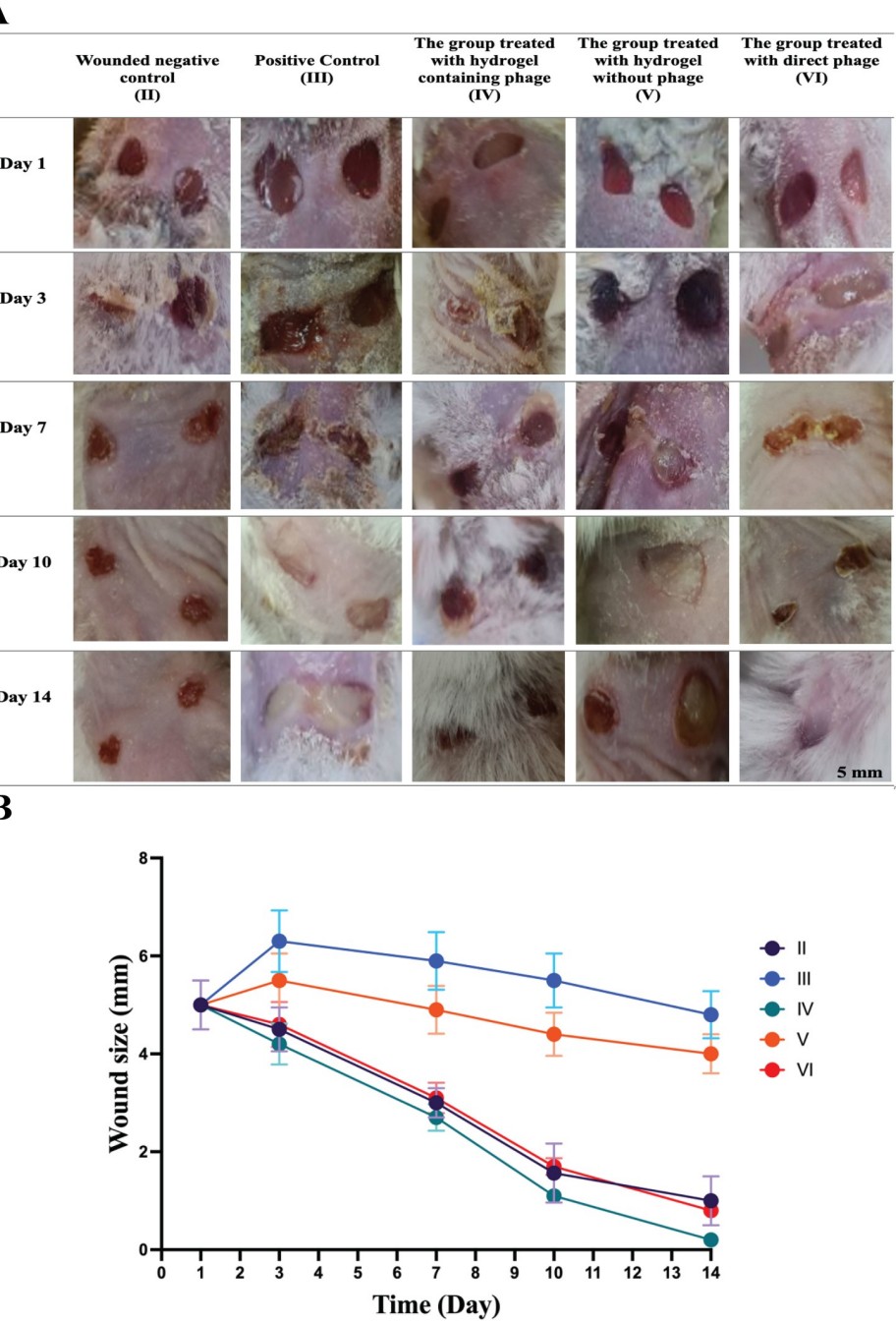

**Fig 10.** (A) Effect of Phage-Containing Hydrogel on Wounds of Male BALB/c Mice Infected with E. faecalis. (II) Negative wound control group (without bacterial infection, no treatment). (III) The positive control (wound, with bacterial infection, and no treatment). (IV) The group was treated with a hydrogel containing bacteriophage (wound infected with E. faecalis). (V) The group was treated with a hydrogel without phage (wound, infected with E. faecalis). (VI) The group treated with direct phage (wound and infected with E. faecalis). (B) Wound healing rates based on wound sizes measured over 14 days in different treatment groups of mice. The graph represents the following groups: II (Wounded negative control): Mice without treatment (blue line). III (Positive control): Mice treated with standard wound healing agents (orange line). IV (Phage-containing hydrogel): Mice treated with hydrogel infused with bacteriophages for enhanced wound healing (gray line). V (Hydrogel without phage): Mice treated with hydrogel alone, without bacteriophages (yellow line). VI (Direct phage): Mice treated with bacteriophages applied directly to the wound (light blue line). All groups' wound sizes decrease over time, with the fastest reduction observed in the Phage-containing hydrogel group (IV) and the slowest in the Wounded negative control group (II). Time points include measurements on days 1, 3, 7, 10, and 14.

**Table 1. Histopathological assessment of tissue parameters collected from wounds on day 14.**

| | Healthy control (I) | Wounded negative control (II) | Positive control (III) | Phage- containing hydrogel (IV) | Hydrogel without phage (V) | Direct phage (VI) |
|---|---|---|---|---|---|---|
| **Density** | 1 | 3 | 3 | 1 | 3 | 3 |
| **Oedema** | 0 | 1 | 2 | 3 | 1 | 1 |
| **Infiltration of inflammatory cells** | 1 | 2 | 3 | 3 | 3 | 3 |
| **Fibrogenesis** | 1 | 3 | 3 | 3 | 3 | 3 |
| **Angiogenesis** | 0 | 1 | 2 | 3 | 1 | 2 |
| **Epithelialization** | 0 | 3 | 0 | 3 | 3 | 2 |
| **Fibrosis** | 0 | 3 | 3 | 0 | 3 | 2 |
| **Hair follicle** | 3 | 0 | 0 | 3 | 0 | 1 |
| **Epidermal thickness** | 3 | 2 | 0 | 3 | 1 | 1 |

(0) no alterations (none), (1) mild alterations (<25%), (2) moderate alterations (25%–50%), and (3) severe alterations (>50%).

inflammatory cell infiltration (1), significant epithelialization (3) (attributed to HA), hair follicle re-formation (3), absence of fibrosis (0), and excellent angiogenesis (3) compared to the positive control. Furthermore, the tissue restoration in the phage-containing hydrogel-treated group closely resembled the control group (healthy, unwounded tissue), which shifted towards restoring the standard skin structure rather than forming fibrotic tissue (scars). The involved cells in the restoration process returned to normal levels consistent with healthy tissue. Notably, this group's proliferation of new capillaries increased significantly, underscoring the phage-containing hydrogel's effect on stimulating angiogenesis in the affected area (Fig 11).

Additionally, there was an acceptable increase in hair follicle bud proliferation, indicating positive effects on hair follicle regeneration and proliferation. Comparing this group with those treated with hydrogel without phage or phage directly revealed that, although repair processes were favorable, tissue repair ultimately led to fibrosis at the wound site, representing a form of wound control rather than natural repair. In contrast, the phage-containing hydrogel-treated group did not form fibrotic tissue and showed no inflammatory cells on day 14 post-wound, indicating normal wound regeneration and repair.

## 4 Discussion

This study presents a hydrogel-based delivery system for administering SAM-E.f 12 phage to treat wounds infected with E. faecalis. This system shows high efficacy and specificity against E. faecalis without virulence factors or antibiotic resistance genes. Hydrogels, known for their biocompatibility and high moisture content, are effective for targeted phage delivery and the prevention of bacterial colonization [34].

In the current study, the hydrogel, composed of SA, CMC, and HA cross-linked with $CaCl_2$, ensured a stable and sustained phage release. Recent advancements have highlighted the effectiveness of hydrogel-based delivery systems in preventing bacterial colonization, particularly in combating MDR infections near femoral tissues [35, 36].

Additionally, gel/hydrogel-based biomaterials have shown great promise in addressing wound healing and infection prevention challenges, as evidenced in several recent studies.

For instance, a study by Feng et al. discusses the use of hydrogel-based platforms in postoperative cancer treatment, highlighting their biocompatibility, adaptive shape, and ability to release drugs in a sustained manner [37]. Although the focus is on cancer treatment, the principles of sustained drug release and antibacterial functionality align closely with the objectives

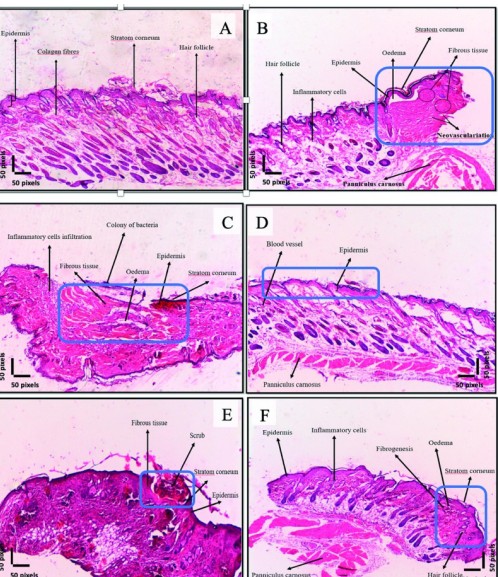

**Fig 11. Histopathological evaluation of wound healing outcomes.** Sections of the skin in different groups with 10x magnifications. The areas marked with blue rectangles indicate the locations of the wounds. (A) I) control group (healthy mice). (B) II) Negative control group (wounded mice, without bacterial infection, without treatment). (C) III) Positive control group (wounded mice, wounds infected by E. faecalis bacteria, without treatment). (D) IV) The group was treated with a hydrogel containing SAM-E.f 12 phage (wounded mice, the wounds were infected with E. faecalis). (E) V) The group was treated with a hydrogel without phage (wounded mice, the wounds were infected with E. faecalis). (F) VI) The group was treated with direct phage (wounded mice, the wounds were infected with E. faecalis).

of our hydrogel system. Similarly, Joorabloo and Liu emphasize the role of reactive oxygen species scavenging in wound healing, underscoring the importance of balancing oxidative stress for tissue regeneration [38]. This is particularly relevant to our study, as ROS scavenging agents could complement phage therapy by promoting skin regeneration while preventing oxidative stress-related tissue damage.

Furthermore, inorganic biomaterials like those discussed by Ma and Wu demonstrate the potential of combining organic and inorganic materials to enhance the bioactivity and stability of hydrogels in skin tissue engineering [39]. While our study focused on organic hydrogel components such as SA, CMC, and HA, incorporating bioactive inorganic particles could further enhance the performance of phage-containing hydrogels in future applications.

In the context of improving contact materials for wound management, Wang et al. provide insights into the importance of understanding the interface chemistry between materials and biological systems [40]. Their findings on the role of metal contacts in optimizing material interfaces for electronic devices can be extrapolated to biological interfaces in wound healing applications. A better understanding of material interfaces may lead to improved integration of hydrogels in biological systems, facilitating more efficient wound healing.

The study by Fu et al. introduces a cyclic heptapeptide-based hydrogel, which demonstrated significant improvements in wound healing for diabetic mice and patients [41]. Combining peptides, nanomaterials, and hydrogels enhances antimicrobial activity and tissue regeneration. This concept is analogous to the dual-action approach we propose in our study, where the hydrogel releases phages to combat bacterial infections and promotes tissue repair through its inherent biocompatibility and healing properties.

By comparing these studies with our findings, we can conclude that integrating multiple polymers into hydrogels, compared to using a single polymer, improves their stability and

physical attributes, making them highly effective as wound dressings to prevent bacterial contamination [42]. Moreover, this study optimized the hydrogel's swelling index and degradation behavior to ensure consistent phage release without compromising structural integrity. These findings align with the research by Liang et al. [43], which developed hydrogels with excellent exudate absorption, moisture retention, and oxygen permeability, further validating the potential of our hydrogel formulation.

This study highlights the potential of phage-containing hydrogels as an innovative solution to antibiotic-resistant wound infections.

This study optimized the hydrogel's swelling index and degradation behavior to ensure the consistent release of phages without compromising structural integrity. The experiments revealed controlled swelling and degradation, ensuring the gradual release of phages and the efficient inhibition of bacterial growth. The hydrogel's swelling behavior aligns with previous reports [21, 36]. Following seven-day immersion, the hydrogel demonstrated a swelling index of 0.43. This enhanced swelling capacity is expected to facilitate the sustained release of phages. However, it is crucial to maintain a balance, as excessive swelling can compromise the hydrogel's structural integrity. Thus, hydrogels with moderate swelling properties may be the most suitable for wound healing applications [44].

The study demonstrated that the phage-containing hydrogel exhibited a gradual weight loss of 23% over seven days. This correlated with the sustained phage release and continuous antibacterial activity observed *in vitro*. This degradation is attributed to the degradable nature of polysaccharides, which are essential for wound healing and are expected to enhance the release of antimicrobial agents as the hydrogel degrades [45].

Similar to other studies, SEM analysis showed that the surface morphology of the phage-containing hydrogel had minor irregularities due to the presence of a phage field [26].

FTIR spectroscopy was used to confirm the chemical composition of the hydrogel, which consists of SA, CMC, and HA. The examination revealed distinct peaks corresponding to each constituent's functional groups. Peaks related to hydroxyl groups were detected in the $3200–3500$ cm$^{-1}$ range, while the $1620–1420$ cm$^{-1}$ range exhibited peaks representing carbonyl and amide groups for SA and CMC. In addition, potential amide groups in HA were identified. Despite some peak overlap, the unique amide peaks characteristic of HA were discernible, indicating the successful integration of HA into the hydrogel structure. This underscores the integrity of the hydrogel formulation and provides insights into its stability and potential applications.

This study demonstrated the efficacy of phage-containing hydrogels in inhibiting bacterial growth using *in vitro* antibacterial activity assays. A steady decrease in the OD of the bacterial culture over time in the presence of the hydrogel indicated its suppressive effect on bacterial proliferation, in contrast to the control group, which showed a significant increase in OD. The disk diffusion assay further supported the sustained release and effectiveness of the encapsulated phage particles, as evidenced by the formation of a halo around the bacteria surrounding the phage hydrogel disk [46].

Comparing these findings with those of a similar study by Kim et al. (2021), confirms the success of hydrogel-based delivery systems in impeding bacterial growth. The consistent outcomes from both studies highlight the potential of these hydrogels as effective strategies for combating antibiotic-resistant bacterial infections and promoting wound healing [36].

The antibacterial properties of the hydrogel were tested in live mice across six different treatment groups to evaluate its effectiveness in reducing bacterial levels in the wounds. By the second day post-wounding, the group treated with the phage-containing hydrogel showed significantly lower bacterial counts than the untreated group. Notably, all mice treated with the hydrogel-embedded phage survived and remained healthy, whereas many mice in other

groups did not survive. Blood cultures from the deceased mice confirmed the presence of *E. faecalis*, indicating fatalities due to bacterial sepsis. These findings align with various research efforts to highlight the efficacy of hydrogel-based treatments for bacterial infections and wound healing [47]. Another study, Field [43], demonstrated the effectiveness of adhesive, antioxidant, and antibacterial self-healing hydrogels for promoting wound healing and fighting infections. A study by Zhao et al. (2020) [48] further supports the idea that hydrogels with rapid shape adaptability and antibacterial properties help address multidrug-resistant bacterial infections. A wound healing study found that phage-containing hydrogels significantly enhanced mouse wound regeneration and antimicrobial activity. The group treated with this hydrogel showed rapid improvement from day one and complete healing by day 14. In comparison, the phage-free hydrogel group experienced delayed healing, and direct phage treatment, although practical, was less successful than the phage-containing hydrogel.

Collectively, these studies emphasize the potential of hydrogel-based systems to deliver antibacterial effects and promote wound healing, consistent with the present study's results. The histological findings, consistent with similar studies [49, 50], demonstrate that phage-containing hydrogel facilitates a more regular and efficient wound-healing process, reducing excessive inflammation and promoting ideal tissue repair. SA-CMC-HA hydrogels loaded with phages exhibit antibacterial properties, stimulate the activity and proliferation of tissue factors related to wound healing, improve skin healing quickly, and serve as an ideal dressing.

However, several challenges remain before phage-containing hydrogels can be widely adopted in clinical settings. The large-scale production of these hydrogels requires standardized protocols to ensure consistency and stability. Potential immune responses to phages must also be thoroughly investigated to prevent adverse reactions. Regulatory approval processes play a crucial role in the clinical implementation of innovative treatments. Exploring the efficacy of phage-containing hydrogels against other bacterial pathogens could further expand their application, making them a versatile tool in the fight against various bacterial infections. Phage-containing hydrogels offer a promising treatment for *E. faecalis*-infected wounds by effectively targeting bacteria and enhancing wound healing through controlled release. This innovative approach addresses antibiotic resistance, promotes tissue regeneration, and provides dual benefits. Future research is needed to standardize production and address clinical challenges for broader applications. The histological findings are comparable with those of similar studies [49, 50] and show that treatment with phage-containing hydrogel helps in a more regular and efficient wound healing process, reducing excessive inflammation and promoting ideal tissue repair. Therefore, besides the antibacterial effect, SA-CMC-HA hydrogels loaded with phages stimulate the activity and proliferation of tissue factors related to wound healing. These can improve skin healing quickly and serve as an ideal dressing.

The potential for bacteria to develop resistance to phages over time is a critical concern in applying phage therapy [51, 52]. Recent studies have highlighted that bacteria can build phage resistance through several mechanisms, including mutations that alter bacterial surface receptors, preventing phage adsorption and subsequent infection, or activating the CRISPR-Cas system, which neutralizes phage DNA [53, 54].

Our research evaluates the efficacy of the phage-containing hydrogel in treating *E. faecalis*-infected wounds. Although this study did not specifically assess the emergence of bacterial resistance to phages, future research should address this concern to ensure the long-term effectiveness of phage therapy. One proposed strategy to combat resistance is using phage cocktails —combinations of different phages targeting various receptors. This approach reduces the likelihood of resistance development, as bacteria must acquire multiple mutations simultaneously to evade the effects of the cocktail [55]. Phage cocktails have demonstrated success in

experimental models, effectively controlling bacterial infections while minimizing the emergence of resistant strains [56].

Additionally, phages' adaptive nature can counteract bacterial resistance. Phages evolve through mutations that enable them to infect resistant bacterial variants, creating an ongoing evolutionary arms race between phages and their bacterial hosts [54]. This co-evolutionary process highlights the importance of understanding bacterial resistance mechanisms and reinforces the potential for phage therapy to remain effective despite developing resistant bacterial populations.

While the primary focus of our study has been to evaluate the phage-containing hydrogel's antimicrobial efficacy, particularly in hydrogel application in wound management, it is also essential to consider the inherent properties of hydrogel components that may contribute to wound healing. HA, a major hydrogel component, is renowned for its healing properties in various clinical scenarios. HA is a naturally occurring polysaccharide found in the extracellular matrix of human tissues, notably in the skin, where it plays a critical role in tissue repair and regeneration [57]. HA's ability to retain moisture and regulate inflammation makes it an excellent facilitator of the wound-healing process [58]. Studies have shown that HA can expedite healing by enhancing epithelialization, reducing inflammation, and promoting angiogenesis [59, 60]. Furthermore, HA's viscoelastic nature helps maintain a moist environment at the wound site, which is conducive to natural tissue regeneration and scar minimization [61]. The integration of HA within our hydrogel is designed to leverage its antimicrobial capabilities and utilize these healing properties, providing a dual-action approach to wound management.

Including HA in the hydrogel supports sustained release of phage and significantly contributes to wound healing. This dual functionality is particularly advantageous as it combines antibacterial and healing properties, which are absent in pure hydrogel treatments. While pure hydrogel exhibits limited antimicrobial action, its HA content does promote some healing, as evidenced by histopathological analysis of treated wounds.

Conclusively, the phage-containing hydrogel not only demonstrates superior performance in reducing bacterial load but also excels in supporting tissue repair. This underscores its potential as a more effective treatment option in phage therapy, further solidifying its value in the field.

## 5 Conclusion

In this study, we developed a novel hydrogel incorporating SAM-E.f 12, a newly characterized lytic phage targeting *E.faecalis*. Using natural polymers like SA, CMC, and HA, and the hydrogel offers optimal wound healing properties, including moisture retention and secretion absorption. The encapsulation of SAM-E.f 12 in this hydrogel allows for a slow, sustained release, enhancing antimicrobial activity and treatment effectiveness.

Our evaluations, conducted in vitro and in vivo using an *E. faecalis*-infected mouse model, demonstrated significant bacterial load reduction and accelerated wound healing. This highlights the hydrogel's potential as a practical and effective solution for treating persistent and chronic infections. We propose that this hydrogel system could be adapted to deliver other phages, broadening its application to various bacterial infections. Future research should focus on optimizing production, ensuring regulatory compliance, and carefully exploring broader applications to harness the potential of phage-containing hydrogels in wounds.

## Acknowledgments

The authors thank the personnel of the Bacteriology Department of the Pasteur Institute of Iran (IPI) for their assistance. I want to express my gratitude to Mr. Mohammad Sholeh for his

invaluable support and guidance throughout the preparation of this article. His profound insights and unwavering encouragement have been instrumental in shaping the direction and depth of this work. I am deeply appreciative of his contributions.

## Author Contributions

**Conceptualization:** Sahar Abed, Masoumeh Beig, Morvarid Shafiei, Abdolrazagh Hashemi Shahraki, Sara Sadeghi.

**Data curation:** Sahar Abed, Morvarid Shafiei.

**Formal analysis:** Sahar Abed, Aria Sohrabi.

**Investigation:** Sahar Abed, Morvarid Shafiei, Abdolrazagh Hashemi Shahraki, Aria Sohrabi.

**Methodology:** Sahar Abed, Seyed Mahmoud Barzi.

**Project administration:** Morvarid Shafiei.

**Software:** Sahar Abed.

**Supervision:** Morvarid Shafiei.

**Validation:** Morvarid Shafiei, Aria Sohrabi.

**Visualization:** Sahar Abed.

**Writing – original draft:** Sahar Abed, Masoumeh Beig.

**Writing – review & editing:** Sahar Abed, Masoumeh Beig, Abdolrazagh Hashemi Shahraki, Sara Sadeghi.

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
