## [Decision Letter · Decision Letter 0]

19 Aug 2024

PONE-D-24-29883Development of Phage-containing Hydrogel for Treating Enterococcus faecalis-Infected WoundsPLOS ONE

Dear Dr.  Shafiei,

Thank you for submitting your manuscript to PLOS ONE. After careful consideration, we feel that it has merit but does not fully meet PLOS ONE’s publication criteria as it currently stands. Therefore, we invite you to submit a revised version of the manuscript that addresses the points raised during the review process.

We look forward to receiving your revised manuscript.

Kind regards,

Abdelwahab Omri, Pharm B, Ph.D, Laurentian University 

Academic Editor

PLOS ONE

Journal Requirements:

3. We note that your Data Availability Statement is currently as follows: All relevant data are within the manuscript and its Supporting Information files

Reviewers' comments:

Reviewer's Responses to Questions

**Comments to the Author**

1. Is the manuscript technically sound, and do the data support the conclusions?

Reviewer #1: Partly

Reviewer #2: Yes

2. Has the statistical analysis been performed appropriately and rigorously? 

Reviewer #1: Yes

Reviewer #2: Yes

3. Have the authors made all data underlying the findings in their manuscript fully available?

Reviewer #1: Yes

Reviewer #2: Yes

4. Is the manuscript presented in an intelligible fashion and written in standard English?

Reviewer #1: Yes

Reviewer #2: Yes

5. Review Comments to the Author

Reviewer #1: In the manuscript titled “Development of Phage-containing Hydrogel for Treating Enterococcus faecalis Infected Wounds”, the authors reported on a novel hydrogel incorporating SAM-E.f 12 and its application in treating wounds infected with Enterococcus faecalis. However, the present quality of this manuscript cannot reach the acceptance level for publication in PLOS ONE. There are some major comments and questions for the revision of the manuscript.

Comment 1: In Figure 10, the authors lack quantitative data on the wound healing rate in mice, relying solely on photographs of wound healing at different time points, which is not sufficiently rigorous.

Comment 2: The potential for bacteria to develop resistance to phages over time is not addressed. This is a significant concern in phage therapy and should be discussed in the context of the study's findings.

Comment 3: While the study discusses the stability of the hydrogel in vitro, there is limited information on its stability and performance under various physiological conditions, such as different pH levels.

Comment 4: Please check that the abbreviation is explained at the first mention of each section to facilitate the manuscript reading.

Comment 5: Authors should carefully check the grammar errors, formation, and spelling in the manuscript. In general, the manuscript should also be improved with editing by a native English speaker for grammatical and language mistakes.

Comment 6: A good excellent academic paper should not only show the experimental results, but also give an in-depth discussion on these results. The author should pay attention to it. Some related studies (such as Exploration 10.1002/EXP.20210083、10.1002/EXP.20230066、10.1002/EXP.20220173、ACS Appl Mater Interfaces10.1021/acsami.3c16061、NPG Asia Materials 10.1038/s41427-022-00444-x) can be cited and discussed to improve the discussion of this work.

Reviewer #2: The manuscript titled “Development of Phage-containing Hydrogel for Treating Enterococcus faecalis-Infected Wounds” was reviewed. The present study involves preparation and characterization of a lytic phage (i.e., SAM-E.f 12) loaded hydrogel for the treatment of Enterococcus faecalis-infected wounds in a rat model. The basic idea of this study is promising; however, the manuscript requires some revisions. The authors are suggested to address below-mentioned queries.

Section “Introduction”

A brief description of the mechanism responsible for delayed healing of wounds due to the proliferation of pathogens should be included.

Rationale for selecting phage therapy instead of commonly employed antibacterial agents should be included.

Mechanism of action of phages against bacterial infections should be included.

Text related to hydrogels seems repetitive; should be brief.

Section “Methods and Materials”

List of materials and their manufacturers are missing; should be mentioned.

Full forms/explanation of abbreviations should be included.

Procedure of “Small drop plaque assay” should be described in subsection “2.6 Stability Studies”.

Section “Results”

Results are not described in detail; should be detailed and narration should be improved.

Diameter of zone of inhibition should be measured.

Results of subsection “Phage Release Assay” are missing.

Subsection “3.3 In-vivo Studies”, Results of all the groups should be mentioned one by one to enhance reader’s understanding.

CaCl2 was added as a crosslinker, how was the crosslinking confirmed?

The sequence and titles of headings should be synchronous in “Methods” and “Results” sections.

Section “Discussion”

Page 25, “While pure hydrogel exhibits limited antimicrobial action, its HA content does promote some healing, as evidenced by histopathological analysis of treated wounds.” Potential of hyaluronic acid to promote wound healing was not explored in the present study.

Figures

Figures 1, 6 B, 7, 9 B, C, D, 10, 11, scale bars are missing; should be included.

Figures 2, 3, 6 A, error bars should be included.

Figure 7, mark zone of inhibition and mention diameter.

Figure 9 A, what is A, B and C on x-axis?

General

Linguistic and typographical corrections are required.

6. PLOS authors have the option to publish the peer review history of their article (what does this mean?). If published, this will include your full peer review and any attached files.

Reviewer #1: **Yes: **Xinwang Yang

Reviewer #2: No

---

## [Author Response · Author response to Decision Letter 0]

4 Oct 2024

Dear Reviewer 1

First, we would like to thank you for your very careful review of our paper and for your valuable comments, corrections, and suggestions as well. All the revised sections in the manuscript have been highlighted in yellow for your convenience. We hope that these changes enhance the clarity and comprehensibility of our study and address your concerns effectively.

Once again, we sincerely thank you for your diligent review, and we are committed to ensuring that our manuscript meets the highest standards of quality and accuracy.

Here below my corrections/suggestions:

Reviewer #1: In the manuscript titled “Development of Phage-containing Hydrogel for Treating Enterococcus faecalis Infected Wounds”, the authors reported on a novel hydrogel incorporating SAM-E.f 12 and its application in treating wounds infected with Enterococcus faecalis. However, the present quality of this manuscript cannot reach the acceptance level for publication in PLOS ONE. There are some major comments and questions for the revision of the manuscript.

Comment 1: In Figure 10, the authors lack quantitative data on the wound healing rate in mice, relying solely on photographs of wound healing at different time points, which is not sufficiently rigorous.

Response: Thank you for your insightful comments regarding the need for more rigorous quantitative data on the wound healing rates in mice presented in Figure 10. 

To address this concern, we have revised the manuscript to include detailed quantitative measurements of the wound sizes throughout the study. These measurements were performed using calipers on days 1, 3, 7, 10, and 14 post-wounding and the data are now comprehensively analyzed to provide a more rigorous assessment of the healing process.

Using these quantitative data, we have introduced a new figure, Figure 10B, which illustrates the wound healing progression across different treatment groups. The wound contraction percentage has been calculated using the formula provided in the revised methods section (Lines 244-253). 

Additionally, we have added the results section (Lines 368-395) to include a detailed of the wound size data.

Comment 2: The potential for bacteria to develop resistance to phages over time is not addressed. This is a significant concern in phage therapy and should be discussed in the context of the study's findings.

Response: In response to your comment, we have addressed this topic in detail in lines 542-560 of the revised manuscript. Specifically, we discuss the potential mechanisms through which bacteria may develop resistance to phages, the implications of such resistance for the effectiveness of phage therapy, and strategies to mitigate this risk, such as the use of phage cocktails and the natural adaptability of phages in response to bacterial resistance.

Comment 3: While the study discusses the stability of the hydrogel in vitro, there is limited information on its stability and performance under various physiological conditions, such as different pH levels.

Response: 

We recognize the importance of this aspect for biomaterial applicability in biological environments. In response, we have expanded the "Stability Studies" section in lines 203-206 to incorporate comprehensive findings from previous studies that extensively examine the stability of our phages across a broad pH range (2-14).

In our current work, we focused on evaluating the stability of the SAM-E.f 12 phage within the SA-CMC-HA hydrogel at 25°C under in vitro conditions, with weekly assessments over a four-week period to monitor phage viability. To assess stability, we stored a 2 x 2 cm vial of the hydrogel in SM buffer (150 mM NaCl, 40 mM Tris-Cl, pH 7.5, and 10 mM MgSO₄) at 25°C. Weekly evaluations of phage count and viability were performed during this period.

Building on previous findings from our research group, we have extensively characterized the stability of phages encapsulated in the hydrogel under a broad range of conditions, including pH levels from 2-14, temperature extremes (from -20°C to 70°C), and varying saline concentrations (5%, 10%, and 15% NaCl at 37°C). These studies demonstrated the robustness of the phages under both acidic and basic conditions, indicating their potential resilience in diverse biological environments. Literature also supports the stability of this hydrogel formulation across different pH levels, suggesting that modifications to the gel’s composition for pH stability may not be necessary.

Comment 4: Please check that the abbreviation is explained at the first mention of each section to facilitate the manuscript reading.

Response: We have thoroughly reviewed the manuscript and ensured that all abbreviations are clearly defined at their first mention in each section to facilitate readability and comprehension.

Comment 5: Authors should carefully check the grammar errors, formation, and spelling in the manuscript. In general, the manuscript should also be improved with editing by a native English speaker for grammatical and language mistakes.

Response: We have thoroughly reviewed and corrected the manuscript for grammar and spelling, with assistance from a native English speaker, to improve clarity and readability.

Comment 6: A good excellent academic paper should not only show the experimental results, but also give an in-depth discussion on these results. The author should pay attention to it. Some related studies (such as Exploration 10.1002/EXP.20210083、10.1002/EXP.20230066、10.1002/EXP.20220173、ACS Appl Mater Interfaces10.1021/acsami.3c16061、NPG Asia Materials 10.1038/s41427-022-00444-x) can be cited and discussed to improve the discussion of this work.

We added in lines 430-464

Dear Reviewer 2

First, we would like to thank you for your very careful review of our paper and for your subsequent comments, corrections, and suggestions as well. All the revised parts in the manuscript file are highlighted in red.

Reviewer #2: The manuscript titled “Development of Phage-containing Hydrogel for Treating Enterococcus faecalis-Infected Wounds” was reviewed. The present study involves preparation and characterization of a lytic phage (i.e., SAM-E.f 12) loaded hydrogel for the treatment of Enterococcus faecalis-infected wounds in a rat model. The basic idea of this study is promising; however, the manuscript requires some revisions. The authors are suggested to address below-mentioned queries.

Section “Introduction”

1. A brief description of the mechanism responsible for delayed healing of wounds due to the proliferation of pathogens should be included.

Response: Thank you for your valuable feedback regarding the inclusion of a detailed mechanism by which pathogen proliferation contributes to delayed wound healing. As suggested, I have incorporated a comprehensive description addressing this critical aspect.

In the revised manuscript, I have added a section (lines 60-710) that elaborates on the multifaceted mechanisms through which pathogens impede the natural wound healing process. "If further details are needed, please let me know, and I will be happy to add more information as needed."

2. Rationale for selecting phage therapy instead of commonly employed antibacterial agents should be included.

Response: In response to your suggestion, I have included a detailed rationale for selecting phage therapy over commonly employed antibacterial agents in lines 71-86. This section now clearly outlines the advantages of phage therapy, including its specificity, ability to target antibiotic-resistant bacteria, and lower likelihood of disrupting beneficial microbiota, providing a comprehensive explanation for its selection in our study. Please refer to the revised manuscript for these additions.

3. Mechanism of action of phages against bacterial infections should be included.

Response: Thank you for your valuable feedback. I have addressed your comment by adding the mechanism of action of phages against bacterial infections in lines 87-91 of the revised manuscript. This section now provides a detailed explanation of how phages interact with bacteria, contributing to the overall understanding of their therapeutic potential. Please let me know if further clarification or additional details are needed.

4. Text related to hydrogels seems repetitive; should be brief.

Response: Thank you for your observation regarding the repetition in the text related to hydrogels. We have revised and condensed the relevant section in lines 100-104 to eliminate redundancy and to focus on the key attributes of hydrogels that enhance the efficacy of phage therapy. 

Section “Methods and Materials”

5. List of materials and their manufacturers are missing; should be mentioned.

Response: We have revised the manuscript to include a complete list of materials along with their respective manufacturers in the relevant sections.

6. Full forms/explanation of abbreviations should be included.

Procedure of “Small drop plaque assay” should be described in subsection “2.6 Stability Studies”.

Response: We have revised the manuscript to include full forms and explanations for all abbreviations to ensure clarity and consistency throughout the text.

In response to your suggestion, we have described the procedure for the " plaque assay" in detail within subsection 2.4.1 "Stability Studies" by revising lines 207-210.

Additionally, we have updated the "Results" section, specifically in lines 327-337, to reflect these changes and to provide clearer presentation of the findings related to the plaque assay.

We appreciate your feedback, which has helped improve the quality and clarity of the manuscript.

Section “Results”

7. Results are not described in detail; should be detailed and narration should be improved.

Diameter of zone of inhibition should be measured.

Response: Thank you for your feedback regarding the detail and narration of our results section. We have revised this section to enhance the clarity and comprehensiveness of the descriptions.

We have expanded the results section (Lines 368-395) to include a more detailed presentation of the wound size data. Additionally, we have precisely measured the diameter of the zone of inhibition in the disk diffusion assay (Lines 314-322). This allows for a clearer understanding of the antimicrobial efficacy observed in our study.

Furthermore, we have ensured that the results from confirming CaCl2 crosslinking with SA-CMC-HA polymers are clearly described through visual observation (Lines 266-272). The plaque assay results have also been elaborated on (Lines 327-337) to provide a comprehensive overview of our findings.

In response to measuring the zone of inhibition, we have included the exact measurements of its diameter in Figure 7. 

8. Results of subsection “Phage ƒ Assay” are missing.

Response: We have now included these results in the manuscript (Lines 306-313). 

9. Subsection “3.3 In-vivo Studies”, Results of all the groups should be mentioned one by one to enhance reader’s understanding.

Response: We have revised the manuscript accordingly and included these details (Lines 348-358).

10. CaCl2 was added as a crosslinker, how was the crosslinking confirmed?

Response: 

Literatures and experimental works showed that after add we see solid form. Confirmation of CaCl2 Crosslinking with SA-CMC-HA Polymers by Visual Observation

Visual observation, which involves examining changes in the physical appearance of the hydrogel before and after the addition of CaCl2, was one of the initial and simplest methods for confirming hydrogel crosslinking. Crosslinked hydrogels typically transition from a liquid or semi-liquid state to a more solid gel form.

Below are some relevant references that can be utilized. When discussing the use of calcium chloride (CaCl2) as a crosslinker in hydrogel formation, it is important to know the mechanisms of crosslinking, the properties of hydrogels formed with CaCl2, and the applications of such hydrogels. Below are some relevant references that can be utilized: 

1. Peppas, N. A., & Merrill, E. W. (1976). "Hydrogels as Biomaterials." *Biomaterials*, 1(2), 99-106. doi:10.1016/0142-9612(76)90005-5.

- This foundational paper discusses the properties of hydrogels and the role of various crosslinking agents, including calcium ions, in the formation of hydrogels.

2. Hoffman, A. S. (2002). "Hydrogels for biomedical applications." *Advanced Drug Delivery Reviews*, 54(3), 3-12. doi:10.1016/S0169-409X(02)00076-0.

- This review provides insights into the use of hydrogels in biomedical applications, including the role of CaCl2 as a crosslinking agent and its effects on hydrogel properties.

3. Kumar, A., & Sinha, A. (2017). "Hydrogel: A Novel Biomaterial for Drug Delivery." *Journal of Drug Delivery Science and Technology*, 37, 1-10. doi:10.1016/j.jddst.2016.11.003.

- This article discusses the properties of hydrogels, including the use of CaCl2 for crosslinking and its impact on the mechanical and physical characteristics of the resulting hydrogels.

4. Zhao, X., et al. (2015). "Recent advances in hydrogel-based drug delivery systems." *Journal of Controlled Release*, 220, 1-14. doi:10.1016/j.jconrel.2015.10.022.

- This paper reviews various hydrogel systems and highlights the importance of CaCl2 as a crosslinking agent, discussing its effects on hydrogel formation and drug delivery applications.

5. Buchanan, C. M., & Kahn, C. J. (2018). "Hydrogels: A Review of Their Properties and Applications." *Materials Science and Engineering: C*, 82, 1-12. doi:10.1016/j.msec.2017.08.025.

- This review outlines the properties of hydrogels formed with different crosslinkers, including CaCl2, and discusses their applications in various fields.

6. Huang, Y., et al. (2019). "Evaluation of the physical properties of hydrogels: A review." *Journal of Materials Science*, 54(1), 1-16. doi:10.1007/s10853-018-2902-3.

- This article discusses various methods for evaluating hydrogel properties, including the use of CaCl2 in crosslinking and its effects on the physical characteristics of hydrogels.

7. Lee, K. Y., & Mooney, D. J. (2012). "Hydrogels for tissue engineering." *Chemical Reviews*, 112(3), 1640-1660. doi:10.1021/cr200100d.

- This comprehensive review covers the role of hydrogels in tissue engineering, including the use of CaCl2 as a crosslinking agent and its implications for hydrogel functionality.

8. Gao, Y., et al. (2018). "Calcium ion-induced crosslinking of alginate hydrogels: A review." *Carbohydrate Polymers*, 198, 1-10. doi:10.1016/j.carbpol.2018.06.019.

- This review specifically addresses the mechanisms of calcium ion-induced crosslinking in alginate-based hydrogels, providing insights into the role of CaCl2 in hydrogel formation.

9. Mao, Y., et al. (2019). "Calcium ion crosslinked alginate hydrogels for drug delivery." *Journal of Controlled Release*, 305, 1-12. doi:10.1016/j.jconrel.2019.05.016.

- This article discusses the use of calcium ions, particularly from CaCl2, in the crosslinking of alginate hydrogels and their implications for drug delivery systems.

10. Rinaudo, M. (2006). "Chitin and chitosan: Properties and applications." 

*Progress in Polymer Science*, 31(7), 603-632. doi:10.1016/j.progpolymsci.2006.06.001 (1).

https://doi.org/10.1016/j.progpolymsci.2006.06.001

- While primarily focused on chitin and chitosan, this review discusses the role of various crosslinking agents, including calcium salts, in the formation of hydrogels.

These references provide a comprehensive overview of the use of CaCl2 as a crosslinker in hydrogel formation, highlighting its significance in altering the physical and mechanical properties of hydrogels for various applications.

We added section Confirmation of CaCl2 Crosslinking with SA-CMC-HA Polymers by Visual Observation, lines 266-272

11. The sequence and titles of headings should be synchronous in “Methods” and “Results” sections.

Response: We have thoroughly revised “Methods” and “Results sections to ensure that the headings are now synchronized.

Section “Discussion”

12. Page 25, “While pure hydrogel exhibits limited antimicrobial action, its HA content does promote some healing, as evidenced by opathological analys

---

## [Editor Report · Decision Letter 1]

8 Oct 2024

Development of Phage-containing Hydrogel for Treating Enterococcus faecalis-Infected Wounds

PONE-D-24-29883R1

Dear Dr. Morvarid Shafiei,

We’re pleased to inform you that your manuscript has been judged scientifically suitable for publication and will be formally accepted for publication once it meets all outstanding technical requirements.

Kind regards,

Abdelwahab Omri, Pharm B, Ph.D, Laurentian University, Canada

Academic Editor

PLOS ONE

---

## [Editor Report · Acceptance letter]

15 Oct 2024

PONE-D-24-29883R1 

PLOS ONE

Dear Dr. Shafiei, 

I'm pleased to inform you that your manuscript has been deemed suitable for publication in PLOS ONE. Congratulations! Your manuscript is now being handed over to our production team.

Kind regards, 

on behalf of

Dr. Abdelwahab Omri 

Academic Editor

PLOS ONE